# Incidence and Predictors of Mortality among Community-Dwelling Older Adults in Malaysia: A 5 Years Longitudinal Study

**DOI:** 10.3390/ijerph19158943

**Published:** 2022-07-22

**Authors:** Yee Xing You, Nurul Fatin Malek Rivan, Devinder Kaur Ajit Singh, Nor Fadilah Rajab, Arimi Fitri Mat Ludin, Normah Che Din, Ai-Vyrn Chin, Michael Fenech, Mohd Zul Amin Kamaruddin, Suzana Shahar

**Affiliations:** 1Dietetics Programme and Centre for Healthy Ageing and Wellness (H-CARE), Faculty of Health Sciences, Universiti Kebangsaan Malaysia, Jalan Raja Muda Abdul Aziz, Kuala Lumpur 50300, Malaysia; youyeexing@ukm.edu.my; 2Nutritional Sciences Programme and Centre for Healthy Ageing and Wellness (H-CARE), Faculty of Health Sciences, Universiti Kebangsaan Malaysia, Jalan Raja Muda Abdul Aziz, Kuala Lumpur 50300, Malaysia; fatinmalek93@gmail.com; 3Physiotherapy Programme and Centre for Healthy Ageing and Wellness (H-CARE), Universiti Kebangsaan Malaysia, Jalan Raja Muda Abdul Aziz, Kuala Lumpur 50300, Malaysia; devinder@ukm.edu.my; 4Biomedical Science Programme and Centre for Healthy Ageing and Wellness (H-CARE), Faculty of Health Sciences, Universiti Kebangsaan Malaysia, Jalan Raja Muda Abdul Aziz, Kuala Lumpur 50300, Malaysia; nfadilah@ukm.edu.my (N.F.R.); arimifitri@ukm.edu.my (A.F.M.L.); 5Health Psychology Programme and Centre of Rehabilitation Science, Faculty of Health Sciences, Universiti Kebangsaan Malaysia, Jalan Raja Muda Abdul Aziz, Kuala Lumpur 50300, Malaysia; normahcd@ukm.edu.my; 6Department of Medicine, Faculty of Medicine, University of Malaya, Kuala Lumpur 50603, Malaysia; avchin@um.edu.my; 7Centre for Healthy Ageing and Wellness (H-CARE), Faculty of Health Sciences, Universiti Kebangsaan Malaysia, Jalan Raja Muda Abdul Aziz, Kuala Lumpur 50300, Malaysia; mf.ghf@outlook.com (M.F.); m.zulamin@ukm.edu.my (M.Z.A.K.)

**Keywords:** community, incidence, mortality, predictors, older adults

## Abstract

With older adults accounting for 10.7% of the Malaysian population, determining the predictors of mortality has now become crucial. Thus, this community-based longitudinal study aimed to investigate the predictors for mortality among community-dwelling older adults using a wide range of factors, including clinical or subclinical. A total of 2322 older adults were interviewed and assessed by trained fieldworkers using validated structured questionnaires. The questionnaire consisted of information on socio-demographic characteristics, health status, neuropsychological and psychosocial functions, lifestyle, dietary intake and biophysical measures. The incidence rate of mortality was 2.9 per 100 person-years. Cox regression analysis indicated that advancing age (Adjusted Hazard Ratio, Adj HR = 1.044, 95% CI: 1.024–1.064), male (Adj HR = 1.937, 95% CI: 1.402–2.675), non-married status (Adj HR = 1.410, 95% CI: 1.078–1.843), smoking (Adj HR = 1.314, 95% CI: 1.004–1.721), a higher fasting blood sugar (Adj HR = 1.075, 95% CI: 1.029–1.166), a lower serum albumin (Adj HR = 0.947, 95% CI: 0.905–0.990), a longer time to complete the TUG test (Adj HR = 1.059, 95% CI: 1.022–1.098), and a lower intake of total dietary fibre (Adj HR = 0.911, 95% CI: 0.873–0.980) were the predictors of mortality in this study. These findings provide an estimated rate of multiethnic mortality in middle-income countries and diet is one of the predictors. These predictors of mortality could be a reference in identifying new public health strategies to ensure longer healthier life spans with lower disability rate among community-dwelling older adults in Malaysia.

## 1. Introduction

The United Nations (UN) classifies people aged 60 years and above as older adults. For the year 2016, the global ageing population for this group was estimated as 12.4%. By the year 2050, it is anticipated that this number will increase to approximately 1.3 billion individuals [1]. Applying the same UN definition, the Malaysian Department of Statistics reported that in 2020, 3.5 million (or 10.7%) of the Malaysian population were in the older adult category [2]. The threats faced by ageing populations, in terms of falling ill, encountering accidents, or death, differ from populations in a younger chronological age category [3]. The increased health risk factors for older adults, make it essential to identify these factors, so that early remedial action can be taken. The accelerating expansion of the elderly population, has led to a growing need to promptly determine the predictors, which could elucidate the mortality risk faced by older adults. This will reduce the incidence of an untimely terminal consequence. According to relevant literature, the mortality predictors among older adults within a stipulated time frame are indisputably age and gender [4]. The predictors for mortality, identified through previous investigations, include issues, such as hospitalisation, capability for performing daily routines, cognitive deficit, lifestyle factors (i.e., indulgence in smoking and lack of a regular exercise routine), depression tolerance, the occurrence of a life-threatening disease, such as cancer, loneliness, lack of support from family members, the individual’s socioeconomic status, and negative self-assessment of health [5,6,7,8].

Luy and Gast [9] conducted a meta-analysis and identified educational level, socioeconomic standing, salary, social network, employment, and health education as the factors contributing to higher death rates among men. Indeed, as early as the middle of the eighteenth, it was established that in terms of an early death, men are at greater risk compared to women [9]. Previous investigations also found that conduct or social issues contribute to higher mortality rates in men, which include smoking, alcoholism, addiction to drugs, laxity with regards to medical needs, lack of knowledge in terms of health issues, a lethargic lifestyle, an irresponsible and risky driving attitude, and a lack of devotion to religion [10,11]. Additionally, men with a childhood entrenched in a family disadvantaged by an inferior socioeconomic standing are at greater risk of suffering cardiovascular related deaths, due to unfortunate experiences, including a lack of family closeness, food deprivation, parents with a smoking habit, and physical or emotional abuse [4,11].

Nevertheless, not many population-based studies have reported mortality risk derived from objectively measured clinical or subclinical parameters, particularly among a multi-ethnic Asian population such as Malaysia. Thus, we aimed to investigate the predictors for mortality among community-dwelling older adults in Malaysia in a 5-years longitudinal study using a wide range of factors, including clinical or subclinical.

## 2. Materials and Methods

### 2.1. Study Design and Respondents

This is a follow-up study of the Longitudinal Study on Neuroprotective Model for Healthy Longevity (LRGS TUA) cohort at the five years endpoint. As shown in Figure 1, a total of 2322 older adults consisting of 1114 men and 1208 women were recruited through a multistage random sampling procedure from four states in Malaysia (representing southern, central, northern, and eastern regions of Malaysia). Respondents were chosen via primary sampling unit (PSU) (selection of state), secondary sampling unit (SSU) (random selection of census circle within the state and tertiary sampling unit (TSU) (selection of living quarters within census circle). Notably, the census circle chosen consisted of at least 10% of older adults. The study protocol was conducted in accordance with the Declaration of Helsinki and approved by the Medical Research and Ethics Committee of the National University of Malaysia (LRGS TUA-NN-060-2013). All respondents provided written informed consent prior to all data collection.

The sample size was calculated based on the number of households that must be selected. The estimation formula [12] for the sample size calculation is presented below:N_h_ = (z^2^)(r)(1 − r)(f)(k)/(p)(n)(e^2^)

N_h_ = the parameter to be calculated and is the sample size in terms of number of households to be selectedz = the statistic that defines the level of confidence desired; usually 95% confidence interval was chosenr = an estimate of a key indicator to be measured by the surveyf = the sample design effect, deff, assumed to be 2.0 (default value)k = a multiplier to account for the anticipated rate of non-responsep = the proportion of the total population accounted for by the target population and upon which the parameter, r, is basedñ = the average household size (number of persons per household)e = the margin of error (MOE) to be attained. MOE was recommended to be set at 10% of r, thus e = 0.1 (r).

Based on the formula, the sample size calculation was reported as following:$$N_{h}=\frac{(z^{2})\left(r\right)\left(1-r\right)\left(f\right)\left(k\right)}{\left(p)(\tilde{n})(e^{2}\right)}\;$$
$$N_{h}=\frac{\left(3.84\right)\;\left(0.50\right)\;\left(0.50\right)\;\left(2\right)\;\left(1.1\right)}{\left(0.084\right)\;\left(4.31\right)\;\left(0.0025\right)}$$
N_h_ = 2322

### 2.2. Data Collection

The mortality data of the LRGS respondents from the year 2014 to 2019 were collected through the vital registration system by the National Registration Department (NRD). LRGS respondents were recruited in March 2014 and follow-up in April 2019. The inclusion criteria of the study were individuals aged 60 years and above with no known dementia or any other psychiatric problems, no severe vision or auditory-related difficulties and minimal functional limitations (not wheelchair-bound or bed-ridden). Exclusion criteria were respondents with MMSE scores below 15, since the respondents are classified as demented individuals [13].

All outcomes were measured at baseline and all respondents were interviewed and assessed by trained fieldworkers using pre-tested structured questionnaires as published earlier [14]. The questionnaire consisted of information on socio-demographic characteristics, health status, neuropsychological and psychosocial functions, lifestyle and dietary intake. Biophysical measures which included anthropometry, blood pressure, physical fitness and functional status were measured. Fieldworkers were fully trained in the data collection procedures and interview skills. Additionally, mortality was recorded at wave IV. The potential mortality predictors considered for this investigation is listed below:

#### 2.2.1. Socio-Demographic Data

Information concerning gender, age, ethnicity, marital status, living arrangement (whether living with others or alone), period of formal education, smoking, alcohol consumption situation, and household income were gathered. Interviews were also conducted to acquire self-reported medical histories.

#### 2.2.2. Nutritional Status and Blood Pressure

Anthropometric measurements were applied to assess the nutritional status. These measurements covered height, weight, waist circumference, mid-upper arm circumference (MUAC), and calf circumference. The standard method was adhered to for these measurements [15]. The body weight in kilograms was divided by the squared standing height in meters, to obtain the body mass index (BMI). The Bio-electrical Impedance Analysis InBody S10 (Biospace, Seoul, Korea), was utilized for the assessment of body composition, while a calibrated digital automatic blood pressure monitor (OMRON, Kyoto, Japan) was employed, to gauge the systolic and diastolic blood pressure.

#### 2.2.3. Biochemical Profiles

The biochemical analysis began by drawing a total of 20 mL from fasting peripheral venous blood, by a qualified phlebotomist using a butterfly syringe. The analysis process focused on fasting blood sugar, total cholesterol, high-density lipoprotein (HDL), low-density lipoprotein (LDL), triglyceride, and albumin.

#### 2.2.4. Cognitive Function Test

The mini-mental state examination (MMSE) were used to assess the global cognitive function [13]; attention and working memory was evaluated using the Wechsler memory scale-revised (WMS-R) for the digit span test [16]; information processing was assessed by way of the digit symbol test [16]; verbal learning and memory were assessed through the Rey auditory verbal learning test (RAVLT) [17] and visuospatial memory was assessed with the application of the visual reproduction test [18].

#### 2.2.5. Physical Performance Test

The assessment for fitness involved the 2-min step test (for durability), the back scratch test (for upper body flexibility), the chair sit and reach test (for lower body flexibility), the chair stand test (for lower body strength), the timed up and go test (TUG) (for balance and mobility), the handgrip test (for upper body strength), and the rapid pace gait speed test [19]. The functional level was assessed through the activity of daily living (ADL) [20] and the instrumental activity of daily living (IADL) [21].

#### 2.2.6. Psychosocial Assessment

The validated Malay version of the geriatric depression scale (GDS-15) [22] was applied to assess possible depression indicators among the respondents with Cronbach alpha of 0.89, while the 12-item adaption of the WHO disability assessment schedule (WHODAS 2.0) was used to determine their disability ranking [23]. The validated Malay version of Medical Outcome Study Social Support Survey (MOSS) questionnaire employed for this investigation, covers four areas: emotional or informational support, tangible support, affectionate support and positive social interaction [24]. The Cronbach alpha of the Malay version of the MOSS questionnaire ranged from 0.616 to 0.902.

#### 2.2.7. Dietary Intake Assessment

While the validated dietary history questionnaire (DHQ) [25] was utilized to assess the dietary intake, the nutritionist pro software (Axxya Systems, Stafford, TX, USA) was employed to analyze the nutrient intake.

### 2.3. Statistical Analysis

All statistical analyses were performed using SPSS version 23.0 (Licensed materials–Property of SPSS. Incorporation an IBM Company Copyright 1989 and 2010 SPSS). Cumulative incidence for mortality was calculated by the number of new mortality cases during the follow-up period divided by the number of respondents at the beginning of the study. The mortality incidence rate was calculated as the number of new mortality cases divided by the total person-time observed between baseline and after five years The age-specific incidence rate of mortality was computed for five-year age categories (60–64, 65–69, 70–74, and 75–79, 80–84 and 85 years and above) using a person-years analysis. The age-specific incidence rate curve was modelled using the Microsoft Excel database. The sociodemographic factors, blood pressure, anthropometric, body composition, physical fitness test, cognitive assessments, biochemical indices, and dietary intake were compared between these two groups (mortality and alive) using a Chi-Square test (X^²^) for categorical variables and independent *t*-test for continuous variables. The results were reported as *n* (%) and the mean ± standard deviation for normally distributed data. The statistical significance value was first set at *p* < 0.05 and adjusted based on the number of tests performed in each table. Starting with univariate analyses to assess the effect of each variable separately, followed by cox proportional-hazards regression. The dependent variable was the mortality status with a survival group as the reference variable (0—survival, 1—death). All covariates (age, sex, marital status and total years of education) were included as control variables in each model. Cox proportional-hazards regression analysis was conducted in two steps. Based on the univariate analysis, all the significant variables were categorised into four different groups according to (1) sociodemographic and medical status; (2) vital signs, anthropometry measurements and health profiles; (3) cognitive function, physical function and psychosocial status; (4) dietary intake. Next, a hierarchical regression analyses were performed for the four categories. The significant variables (*p* < 0.05) in each regression model were selected to enter the final cox regression model. The adjusted hazard ratios (HRs) and their 95% confidence intervals (CIs) were estimated with significant value set at *p* < 0.05. Significant variables in the final model were accepted as the predictors of mortality among respondents.

## 3. Results

Of the 2322 respondents in this study, a total of 1986 (85.6%) respondents were alive and 336 (14.4%) were reported dead after five years of follow-up. Table 1 presents the cause of death of the respondents, where most of them died due to ageing-related sickness (43.8%).

The five-year cumulative incidence of mortality was 14.4%. The observed incidence rate of mortality within the five years period was 2.9 per 100 person-years. Figure 2 depicts the estimated age-specific incidence rate of mortality from the baseline to the five years follow-up. In the age group of 60–64 years, the rates increased with age from 1.5 per 100 person-years to 1.9 per 100 person-years in the 65–69 years age group. Then, the rates substantially increased to 3.2 per 100 person-years in the 70–74 years age group, to 5.3 per 100 person-years in the 75–79 years age group, 6.3 per 100 person-years for 80–84 age group and 10.6 per 100 person-years in the age group of 85 years and above. The incidence rate of mortality was estimated to increase two-fold with an increase in age by 10 years.

Table 2 presents the respondents’ baseline characteristics who were alive and those who died after five years of follow up. In comparison to respondents who were alive after five years, those who died were found to be older, male, Malay, non-married, living alone, having lower mean years of education, having a lower household income, smokers, diagnosed with type 2 Diabetes Mellitus, stroke, heart disease and gout (*p* < 0005).

As shown in Table 3, the systolic blood pressure, waist-hip ratio, fasting blood sugar were significantly higher, while the weight, body mass index, mid-upper arm circumference, calf circumference, hip circumference, skeletal muscle mass and serum albumin were significantly lower among respondents who died earlier as compared to those who were alive (*p* < 0.003).

Respondents who had a lower physical fitness (chair sit and reach, 2-min step, chair stand, timed-up-and-go (TUG) and back scratch tests) and IADL score and a poorer cognitive assessments (MMSE, Digit span, Rey Auditory Verbal Learning Test (RAVLT), Digit Symbol, VR I, and VR II) had died as compared to those who were alive after five years (*p* < 0.003) (Table 4).

With regards to the dietary intake (Table 5), almost all respondents did not meet the Malaysian recommended nutrient intake for total fibre, vitamin K, niacin, pyridoxine, folate, calcium, zinc, copper and magnesium intake. Energy intake appeared to be significantly lower in respondents who died as compared to those who were alive after five years (*p* < 0.05). Furthermore, total fibre, vitamin C, vitamin D, riboflavin, niacin, folate, iron, selenium, copper, and magnesium were significantly lower among respondents who died as compared to those who were alive after five years (*p* < 0.002). Whilst, the dietary intake of the respondents based on gender was presented in Appendix A.

As displayed in Table 6, the results from the Cox regression analysis indicated that smoking (Adj HR = 1.314, 95% CI: 1.004–1.721, *p* < 0.05), a higher fasting blood sugar (Adj HR = 1.075, 95% CI: 1.029–1.166, *p* < 0.01), a lower serum albumin (Adj HR = 0.947, 95% CI: 0.905–0.990, *p* < 0.01), a longer time to complete the TUG test (Adj HR = 1.059, 95% CI: 1.022–1.098, *p* < 0.001), and low intake of total fibre (Adj HR = 0.911, 95% CI: 0.873–0.980, *p* < 0.01) were the predictors of mortality in this study. Notably, after adjusting the confounding variables, the odds ratio of all potential variables in predicting mortality was reduced but remained as significant predictors of the mortality incidence. The receiver operating characteristic (ROC) curves with the area under the curve score (AUC = 76.7%) reflected the accuracy of the final model with good sensitivity (85.8%) and specificity (85.5%) for predicting mortality among community-dwelling older adults.

The survival functions of different variable grouping factors of the respondents are illustrated in Figure 3a–e.

## 4. Discussion

This study successfully estimated the incidence rate of mortality of 2.9 per 100 person-years among older adults in Malaysia. In other words, for every 100 respondents involved in this study, it is estimated that three older adults will die per year, which is slightly higher than the mortality rate reported among Singaporean older adults (2.67 per 100 person-years) [26]. It is noteworthy that the studies on the incidence rate of mortality are not widely reported, particularly among multi-ethnic Malaysian older adults. A cohort study conducted among older adults in the United Kingdom reported a higher all-cause mortality rate with 11 per 100 person-years for fallers and 16.8 per 100 person-years for recurrent falls after three years [27]. On the contrary, a China cohort study found that the mortality incidence rate was 0.7 per 100 person-years among Chinese older adults after 12-years follow-up [28], which is lower in comparison to the present result. The variation in the incidence rate might be influenced by the geographic settings, follow-up time, sample size and different health diagnoses considered among the studied population. An increasing age-adjusted trend in mortality was also observed among Malaysian older adults, probably associated with the pace of ageing considering that physiological deterioration increases as people age [29].

In terms of the risk of death, age is frequently deemed a compelling predictor. This could be attributed to the fact that the probability of contracting a lingering ailment, or succumbing to incapacitation, increases in tandem with ageing [30]. This is in agreement with the results from our investigation, which indicate that the probability of mortality, increases with each passing year among Malaysian older adults. Similarly, functional capacity also fades with time, thus escalating the vulnerability to health issues, and the likelihood of death [4]. Reliance on assistance, when it comes to the performance of activities of daily living (such as food consumption, cleaning up, getting dressed, and toilet visits), also increases with age. To compound matters, ageing also increases the incidence of chronic health issues, which include hypertension, diabetes and osteoarticular diseases [31,32]. The data derived from this study revealed that the mortality incidence rate can be anticipated to double with each 10-year increase in age. This translates into a 50% increase in the risk of death after 10 years. In the context of developed countries, the risk of death rises by 50% for every five-year increase in age [4,33].

Smoking-related ailments (including cardiovascular diseases and cancer), have long been identified, as significant mortality risk factors. A systematic review by Müezzinler et al. [34] compiled the results from 22 studies to report joint mortality of >100% and more than one-third, for current and former smokers, respectively, compared to those who never indulged such habits. In the context of mortality advancement, these rates can be interpreted as 6.7 and 2.6 years for current and former smokers, respectively [35]. Efforts to reduce smoking and promote smoking cessation even at an older age are likely to bear a major public health impact [36]. Thus, it is clear, that the effects of smoking are undoubtedly a major mortality risk factor, for those aged 60 and above. Given the health hazards associated to smoking, serious efforts ought to be in the pipeline, to significantly curb smoking habits in all age groups to substantially reduce the morbidity and mortality associated with this indulgence.

Multiple potential risk factors explain the high mortality observed in older adults with high fasting blood sugar. A high glucose reading among older adults, particularly those afflicted with diabetes mellitus, raises the mortality risk considerably [37,38]. Accordingly, our investigations revealed that for older adults with diabetes, the risk of mortality can increase by 1.293. It is apparent, that the pathophysiological impairment related to chronic hyperglycaemia, stemming from the glucotoxic and lipotoxic setting, which come together in diabetes, is the main instigator of mortality [39]. This situation recurs for the cause of death among individuals with chronic hyperglycaemia, with cardiovascular disease being the most prevalent [40]. Previous studies have revealed the association between chronic hyperglycaemia and the increased risk of end-organ problems, such as retinopathy and kidney disease [41]. These events emphasise the progressive effects on microvascular dysfunction, which could lead to an increased mortality risk among older adults saddled with long-term diabetes [39]. The Clinical Practice Guidelines for Malaysian older adults, offers general diabetic management recommendations, which are imperative for slowing down the advancement of macrovascular and microvascular complications that might ultimately culminate in mortality [42].

Additionally, serum albumin was one of the mortality predictors for older adults in this study. A drop in 1 g/L of serum albumin raises the probability of mortality occurrence by 7.4%. A predictive warning for the onset of disability and mortality is reflected when serum albumin decreases in older adults either in a communal or medical setting. A previous study reported that hypoalbuminemia was associated with 70% mortality risk, and this risk remained unchanged for those without normalisation of albumin. Meanwhile, the normalisation of albumin levels was associated with lower mortality risk (51%) [43]. Hypoalbuminemia was associated with long-term chronic malnutrition in older adults which was caused by a decline in food consumption [43]. A decrease in the sense of smell and taste, fluctuations in the hormones that control gastric and intestinal motility, as well as alterations in temperament (including depression, loneliness and dementia) were the most significant sources of geriatric anorexia [44,45]. While the link between serum albumin and mortality calls for thorough investigations, it is important that the clinical decisions arrived at, are not confined to any solitary parameter. Thus, the deficiency in serum albumin can be rectified through an appropriate nutritional intervention, which will consequently decrease the occurrence of mortality among older adults.

According to the results, from the timed-up and go (TUG) test we conducted, those with a poor TUG performance face an increased susceptibility to mortality. This is in agreement with the results attained through previous studies in this area [46,47,48]. It was observed that a single-unit climb, in the TUG test, raised the probability of mortality substantially. The TUG test is recommended as a single measurement procedure, to distinguish older adults prone to life-threatening consequences [49]. The comorbidity burden among older adults, identified through the TUG test, emphasizes the importance of the various interactions among several diverse body systems (this includes the nervous, cardio-pulmonary and musculoskeletal system), with regards to synchronisation of movement and stability [50,51]. A reduced gait pace, for instance, is an indication of sub-clinical cerebrovascular disease, even among high-level performing older adults [48,51]. Walking speed, a crucial component of TUG, is taken into consideration, to identify fundamental biological changes, as well as analysed and unanalysed ailments [52]. Walking speed is also a measurement of energy level, as the act of walking, exerts strains on the performance and structures of the body [53].

In terms of dietary intake, we discovered that the respondents did not fulfil the Recommended Nutrient Intake [54] for total dietary fibre, vitamin K, niacin, pyridoxine, folate, calcium, zinc, copper and magnesium, as prescribed through previous studies, conducted by Malaysian researchers [54,55]. We observed that a decrease in fibre consumption, by 1 g, raises the risk of mortality incidence by 8.9%. The results from our investigations revealed that the total fibre intake, of older respondents, is substantially below the recommended level of 20 to 30 g/day [56]. Fibre plays a significant role in reducing the mortality incidence by: (a) lowering the cholesterol level, (b) bringing down the blood pressure level, (c) sensitising insulin, (d) generally improving glycaemic management, and (e) inducing anti-inflammatory properties [57,58]. These biological effects, which can be linked to the reducing of risk with regards to several prominent chronic ailments (including cardiovascular diseases and cancer), serve as protection against the incidence of mortality [59]. Furthermore, an increase in the intake of soluble and insoluble fibre lessens the risk of all-cause mortality [60].

There are several strengths of the current study. First, this study uses numerous of parameters to measure anthropometry, cognitive function, psychosocial status, physical function and dietary intake as possible predictors for mortality in older adults. Second, this study involved a five years longitudinal study from four different regions in Malaysia, comprising a big sample size population and a good representation of the Malaysian older population. However, this study also has several limitations. The medical illness of the respondents was obtained using self-reported questionnaire. Although in the presence of such limitations, the findings may be useful in some way. These data are useful to understand the causes of premature death which is an important step toward the design of future research and public health policy in this field. Malaysian food composition in nutritionist pro is incomplete in terms of vitamin D, E, K, folate, selenium, copper, and magnesium. Hence, the results of this study can make a better health care policy and intervention for the ageing population. It is useful for healthcare professionals and caregivers for the early detection of disease risk and health status in older adults.

## 5. Conclusions

In conclusion, the incidence rates of mortality among the community-dwelling older adults population in Malaysia were 2.9 per 100 person-years at a 5 years follow-up. Increased age, male gender, non-married status, high fasting blood sugar, low serum albumin, lower performance in the TUG test and low dietary fibre intake were predictors in the mortality among Malaysian older adults. Our findings could be a reference to find and develop new and better strategies to ensure a longer and healthier life among community-dwelling older adults in Malaysia. For example, mortality risk indicators such as sociodemographic and dietary factors could be used to identify the high-risk individuals and educational points for lifestyle changes towards healthy ageing. Although smoking has been proven to be hazardous and increased mortality risk, these findings can strengthen Malaysia’s smoking cessation strategies policy. It is recommended for Malaysians to adhere to healthy lifestyles such as non-smoking, healthy eating, and being physically active to control vascular-related comorbidities.

## Figures and Tables

**Figure 1 ijerph-19-08943-f001:**
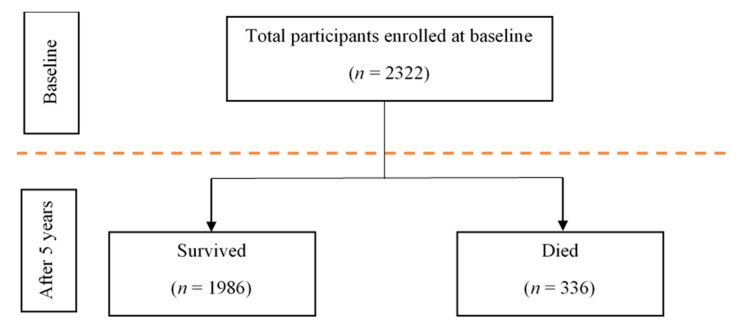
Illustration of the number of respondents at baseline and after five years follow up.

**Figure 2 ijerph-19-08943-f002:**
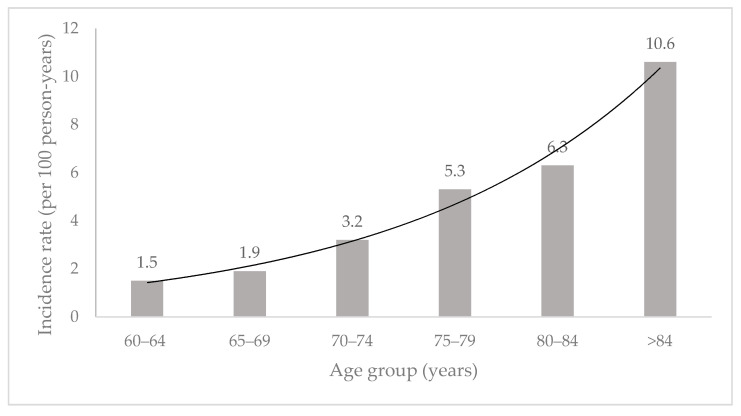
The age-specific incidence rates and 95% confidence intervals of mortality events from 2013 to 2019. The incidence rate was computed for 5 age categories using a person-years analysis and was plotted at the age group: 60–64 years, 65–69 years, 70–74 years, 75–79 years, 80–84 years, and 85 years and above.

**Figure 3 ijerph-19-08943-f003:**
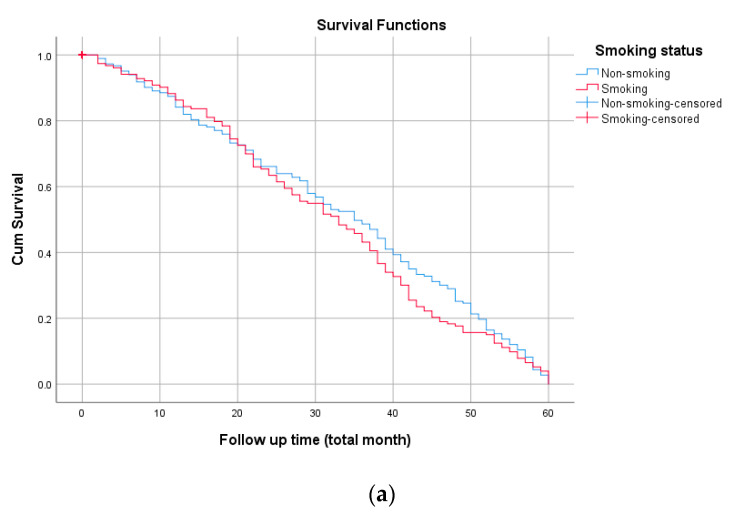
The plot of the estimate of Kaplan-Meier survivor curves of respondents. (**a**) Smoking status, (**b**) fasting blood sugar, (**c**) serum albumin, (**d**) TUG test, and (**e**) fiber intake.

**Table 1 ijerph-19-08943-t001:** The cause of death.

Cause of Death	No. of Respondents	Percentage (%)
Old sickness	147	43.8
Heart disease	47	13.9
Sepsis	40	11.9
Pulmonary disease	32	9.5
Cancer	28	8.3
Stroke	12	3.5
Trauma	10	3.0
Type 2 Diabetes Mellitus	7	2.1
Hypertension	4	1.2
Multiorgan failure	3	1.0
Kidney disease	2	0.6
Gastrointestinal disease	2	0.6
Liver disease	1	0.3
Fever	1	0.3
Total	336	100.0

**Table 2 ijerph-19-08943-t002:** The baseline characteristic of *p* respondents who were alive or died after five years [presented as mean ± standard deviation (sd) or *n* (%)].

Parameters	Alive (*n* = 1986)	Died (*n* = 336)	*p*-Value
Age (years)	68.44 ± 5.88	72.63 ± 7.03	<0.001 *
Gender			<0.001 *
Men	909 (45.8)	205 (61.0)	
Women	1077 (54.2)	131 (39.0)	
Ethnicity			<0.001 *
Malay	1194 (60.1)	253 (75.3)	
Chinese	681 (34.3)	69 (20.5)	
Indian	106 (5.3)	14 (4.2)	
Marital status			<0.001 *
Single/widowed/divorced	598 (30.1)	139 (41.4)	
Married	1388 (69.9)	197 (58.6)	
Years of education	5.34 ± 4.04	3.93 ± 3.40	<0.001 *
Household income (RM or USD)	1368.49 ± 249.57 or 330.75 ± 60.32	965.08 ± 351.06 or 233.25 ± 84.85	<0.001 *
Living arrangement			0.046
Living alone	200 (10.1)	46 (13.7)	
Living with others	1786 (89.9)	290 (86.3)	
Smoking			<0.001 *
Yes	539 (27.1)	153 (45.5)	
No	1447 (72.9)	183 (54.5)	
Alcohol drinker			0.157
Yes	86 (4.3)	9 (2.7)	
No	1900 (95.7)	327 (97.3)	
Hypertension			0.460
Yes	912 (45.9)	147 (43.8)	
No	1074 (54.1)	189 (56.2)	
Type 2 Diabetes Mellitus			0.027 *
Yes	457 (23.0)	96 (28.6)	
No	1529 (77.0)	240 (71.4)	
Hyperlipidaemia			0.491
Yes	514 (25.9)	81 (24.1)	
No	1472 (74.1)	255 (75.9)	
Stroke			0.043
Yes	21 (1.1)	8 (2.4)	
No	1965 (98.9)	328 (97.6)	
Heart disease			0.004
Yes	160 (8.1)	43 (12.8)	
No	1826 (91.9)	293 (87.2)	
Chronic kidney disease			0.488
Yes	26 (1.3)	6 (1.8)	
No	1960 (98.7)	330 (98.2)	
Gout			0.011
Yes	63 (3.2)	20 (6.0)	
No	1923 (96.8)	316 (94.0)	

* Significant at *p* < 0.003,using Independent *t*-test for continuous variables and Chi-square test for categorical variables.

**Table 3 ijerph-19-08943-t003:** The baseline vital signs, anthropometry measurements and health profiles of respondents who were alive or died after five years [presented as mean ± standard deviation (sd)].

Parameters	Alive (*n* = 1986)	Died (*n* = 336)	*p*-Value
Systolic blood pressure (mmHg)	140.84 ± 21.58	145.15 ± 23.19	0.001 *
Diastolic blood pressure (mmHg)	77.31 ± 12.85	78.00 ± 13.80	0.393
Height (cm)	155.88 ± 8.63	155.58 ± 8.84	0.557
Weight (kg)	61.06 ± 12.19	59.44 ± 12.55	0.040
Body mass index (kg/m^2^)	25.08 ± 4.34	24.46 ± 4.83	0.030
Mid-upper arm circumference (cm)	28.54 ± 3.38	27.66 ± 3.96	<0.001 *
Calf circumference (cm)	33.45 ± 3.75	32.15 ± 4.10	<0.001 *
Waist circumference (cm)	88.23 ± 11.09	88.37 ± 12.28	0.837
Hip circumference (cm)	96.78 ± 9.34	94.76 ± 10.41	<0.001 *
Waist-hip ratio	0.91 ± 0.07	0.93 ± 0.07	<0.001 *
Body fat percentage (%)	39.23 ± 10.30	38.56 ± 9.66	0.264
Skeletal muscle mass (kg)	19.41 ± 4.65	18.86 ± 4.74	0.044
Fasting blood sugar (mmol/L)	6.12 ± 1.86	6.59 ± 2.53	<0.001 *
Total cholesterol (mmol/L)	5.41 ± 1.00	5.46 ± 1.06	0.470
HDL cholesterol (mmol/L)	1.40 ± 0.31	1.37 ± 0.33	0.131
LDL cholesterol (mmol/L)	3.32 ± 0.90	3.38 ± 0.97	0.259
Triglyceride (mmol/L)	1.51 ± 0.69	1.55 ± 0.67	0.309
Albumin (g/L)	42.89 ± 2.38	42.35 ± 2.68	<0.001 *

* Significant at *p* < 0.003 using Independent *t*-test.

**Table 4 ijerph-19-08943-t004:** The baseline cognitive function, physical function and psychosocial status of respondents who were alive or died [presented as mean ± standard deviation (sd)].

Parameters	Alive (*n* = 1986)	Died (*n* = 336)	*p*-Value
Cognitive tests			
Mini mental State examination	23.13 ± 4.68	20.89 ± 5.51	<0.001 *
Digit Span	7.55 ± 2.39	7.07 ± 2.52	0.001 *
RAVLT	37.88 ± 10.05	33.79 ± 8.87	<0.001 *
Digit Symbol	5.04 ± 2.46	4.24 ± 1.65	<0.001 *
Visual Reproduction 1	44.01 ± 32.18	35.73 ± 29.36	<0.001 *
Visual Reproduction 2	36.17 ± 34.25	26.83 ± 29.60	<0.001 *
Fitness tests			
Chair sit and reach test	1.27 ± 11.31	3.58 ± 12.92	0.002 *
2-min step test	62.03 ± 25.01	51.42 ± 25.07	<0.001 *
Chair stand test	9.94 ± 2.93	8.47 ± 2.83	<0.001 *
TUG test	10.73 ± 3.08	12.77 ± 3.57	<0.001 *
Back scratch test	14.60 ± 12.35	19.66 ± 12.22	<0.001 *
Psychosocial and functional status			
Activities of daily livings (ADL)	6.00 ± 0.02	5.98 ± 0.33	0.262
Instrumental activities of daily living (IADL)	12.52 ± 2.18	11.35 ± 3.14	<0.001 *
Geriatric depression scale	2.67 ± 2.24	2.87 ± 2.28	0.144
WHO Disability Assessment Schedule (WHODAS)	7.68 ± 10.04	7.54 ± 9.65	0.820
Medical Outcome Study Social Support Survey (MOSS)	39.48 ± 14.40	39.88 ± 15.05	0.641

* Significant at *p* < 0.003 using Independent *t*-test.

**Table 5 ijerph-19-08943-t005:** The baseline dietary nutrient intake of respondents who were alive and those who died after 5 years [presented as mean ± standard deviation (sd)].

Nutrients at Baseline	Alive (*n* = 1986)	Died (*n* = 336)	RNI 2017	*p*-Value
Energy (kcal)	1653.78 ± 462.95	1592.25 ± 479.47	1550–1780	0.025 *
Carbohydrates (g/day)	224.05 ± 74.68	216.79 ± 74.49		0.099
Protein (g/day)	70.55 ± 21.40	68.17 ± 21.42	50–58	0.060
Fat (g/day)	52.60 ± 19.00	51.14 ± 24.67		0.304
Total fibre (g/day)	3.98 ± 2.44	3.21 ± 1.94	20–30	<0.001 *
Vitamin A (RE/day)	1196.29 ± 781.68	1163.00 ± 877.75	600	0.514
Vitamin C (mg/day)	117.26 ± 81.01	98.24 ± 73.19	70	<0.001 *
Vitamin D (mg/day)	0.34 ± 2.14	0.24 ± 1.01	0.015–0.02	0.418
Vitamin E (mg/day)	11.03 ± 5.43	8.63 ± 5.30	7.5–10.0	0.452
Vitamin K (mg/day)	18.32 ± 6.70	11.45 ± 3.94	55–65	0.068
Thiamin (mg/day)	1.54 ± 3.52	1.36 ± 3.15	1.1–1.2	0.366
Riboflavin (mg/day)	1.22 ± 0.48	1.15 ± 0.49	1.1–1.3	0.008
Niacin (mg/day)	10.33 ± 3.88	9.83 ± 3.66	14–16	0.027 *
Cobalamine (μg/day)	3.87 ± 3.56	3.89 ± 3.19	4.0	0.925
Pyridoxine (mg/day)	0.70 ± 0.35	0.68 ± 0.38	1.5–1.7	0.237
Folate (µg/day)	105.53 ± 72.72	93.56 ± 66.02	400	0.005
Calcium (mg/day)	515.98 ± 236.39	493.06 ± 252.56	1000	0.104
Iron (mg/day)	13.41 ± 5.19	12.79 ± 5.63	11–14	0.044
Selenium (µg/day)	24.23 ± 18.08	20.36 ± 16.11	23	<0.001 *
Zinc (mg/day)	3.60 ± 1.86	3.53 ± 2.31	4.4	0.526
Copper (mg/day)	0.58 ± 0.33	0.53 ± 0.33	0.9	0.006
Magnesium (mg/day)	130.79 ± 63.51	120.49 ± 59.57	420	0.006

* Significant at *p* < 0.002 using Independent *t*-test.

**Table 6 ijerph-19-08943-t006:** Potential predictors for mortality at five years.

Predictors of Interest	B	Adj HR (95% CI)	*p*-Value
Smoking, yes	0.273	1.314 (1.004–1.721)	0.047 *
Presence of diabetes	0.257	1.293 (0.999–1.674)	0.051
Fasting blood sugar	0.072	1.075 (1.029–1.166)	0.001 *
Systolic blood pressure	0.004	1.004 (0.999–1.009)	0.106
Mid-upper arm circumference	−0.020	0.980 (0.947–1.015)	0.261
Waist-hip ratio	1.418	4.128 (0.817–20.860)	0.086
Serum albumin	−0.055	0.947 (0.905–0.990)	0.017 *
Mini-mental State Examination	−0.010	0.990 (0.964–1.016)	0.448
RAVLT (immediate recall)	−0.005	0.995 (0.981–1.009)	0.495
Chair stand test	−0.046	0.955 (0.911–1.001)	0.055
TUG test	0.058	1.059 (1.022–1.098)	0.002 *
Instrumental Activity Daily Living (IADL)	−0.018	0.983 (0.941–1.026)	0.426
Total fibre intake	−0.078	0.911 (0.873–0.980)	0.008 *

* Significant at *p* < 0.05 after adjusted for age, gender, marital status and years of education using forward-stepwise Cox Regression analysis. TUG, timed-up-and-go test; ref, reference group; Adj HR, adjusted hazard ratio.

## Data Availability

The datasets used and/or analysed during the current study are available from the corresponding author on reasonable request.

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
