# Peer review of "Incidence and Predictors of Mortality among Community-Dwelling Older Adults in Malaysia: A 5 Years Longitudinal Study"

_ijerph, 2022, doi:10.3390/ijerph19158943_

Round 1
Reviewer 1 Report
Report on Incidence and predictors of mortality among community-dwelling older adults in Malaysia: A 5 years Longitudinal study (ijerph-1753619)
Based on the aims and scope of the journal, an in-depth revision of the language writing, applications, and conclusions of the paper is needed before possible publication.
As a general comment, this paper investigates the predictors for mortality among community-dwelling older adults in Malaysia in a 5-years 80 longitudinal study using a wide range of factors, including clinical or subclinical
First, the authors should state two crucial points in the Abstract and the Introduction of the paper:
1. The general aim of the paper should be clearly stated in the abstract
2. The main contributions of the paper.
Specific comments
Pag 4, the Authors should name the two groups, dead and alive.
Pag 4, Why do authors check each variable separately? They should justify the two steps in Cox regression. As far as I know, they can fit all the variables and then a stepwise selection which is quicker and more efficient.
Pag 5, the sample is too small to study the cause of death and obtain robust conclusions.
Pag 6-9, Tables 2-6, the sample is too small to obtain a robust model fit; some categories could have just a few observations. Authors should present a descriptive analysis of variables, including the size of each category. Are there any missing values?
Pag 9, the Authors should explain the ROC curve and applied explicitly for survival analysis.
Pag 10, Discussion, Authors should explain which population they could extend their results. For example, in which population? "In terms of gender, men are 1.937 times more vulnerable than women, when it comes 306 to the subject of mortality risk."
Pag 10, Authors should unify bibliography style numbered or author-year, for example, "On the contrary, Zhu et al. (2019) reported that the mortality incidence rate was 658.50 per 100,000 person-years among Chinese older adults after 285 12-years follow-up [25]".
Pag 13, How do authors propose to use their results to ensure a longer and healthier life among community-dwelling older adults in Malaysia? Which strategies?
Reviewer 2 Report
The paper titled "Incidence and predictors of mortality among community-dwelling older adults in Malaysia: A 5 years Longitudinal study" aims at determining the mortality incidence among community-dwelling older adults in Malaysia using a community-based longitudinal study. In addition to the incidence, the current manuscript also tries to determine the predictors of overall survival. My specific comments with respect to the paper is provided below:
1) Unclear from the data collection section as to how often the NRD for vital statistics updated. So for instance when will the vital statistics for respondents in 2019 were updated.
2) Discuss the different tools used in the analysis, in particular the MMSE score. Why it is used and what does the score of MMSE refers to. As it is mentioned that patients with MMSE score < 15 were not included. Provide some context for the readers.
3) Which validated structured questionnaires are being discussed in the manuscript?
4) Details are not provided as to how the time to survival was calculated. What was used as the start date and end date used for the calculations? How was time for censored events calculated?
5) All the different variables like nutritional status, blood pressure, biochemical profiles, cognitive function test etc. when will these measured. Was is at baseline or other time points on an ongoing basis. Needs to be clearly mentioned in the manuscript.
6) For the tools used for the psychosocial assessment, provide the Cronbach's alpha and reliability tests.
7) Were there any respondents who were lost to follow-up?
8) The steps used for the Cox regression analysis is unclear. Which variables were included in the univariate analysis and how was it decided to include or exclude variables in the model. What is the meant by "all significant variables in the univariate analysis were categorized into few different groups..."? Very confusing and hard to understand. What was the p-value chosen for the univariate analysis?
9) Given the number of variables included in the analysis, was the study powered enough?
10) Which software was used to conduct the study analysis?
11) What was the median follow-up for the study population?
12) In table 2, the proportions presented in the study is incorrect, as you are comparing the survived vs. deceased the column percentage should be presented. Smoking variable in the table needs to be fixed. The footer for the table needs to include only Chi-square tests.
13) Also for table2, 3 4 and 5 so many comparisons are being conducted, why no multiple comparison tests were used for the p-values? For table 5, the dietary nutrient intake for some of the variables would be different for males and females. Did you find any gender difference in the intake variables?
14) For table 6, provide clearly what are the HR being adjusted for. Also in the text it was mentioned that age, sex and years of education were adjusted for but in the table you still include age and gender. Can you explain what was the need for presenting those variables. And if you including them in the model then just say adjusted for years of education.
15) For table 6, please exclude the parameter estimates column. The HR should be presented in the same row as the variable of interest and not with the generic variable name. Very difficult to read the table.
16) Was multicollinearity checked for the variables included in the final multivariate model? If you see the waist-hip ratio variable which is continuous I believe but has really very high HR's. Also in the discussion section it is briefly mentioned about the serum albumin and fasting blood sugar level are correlated. Why were both of these included together in the model, this can result in multicollinearity.
17) For the mortality rate comparison, it would be advisable to have same unit for the person years, it makes the comparison easier for the reader.
18) Line 306- 321, needs to be changed. As there it is mentioned that Male are 1.937 times more likely to die but from the table 6, males are the reference and females are at higher risk of mortality. So this paragraph needs to be revisited.
19) For the conclusion section, based on the results of your data analysis what would be the recommendation? Expanding on this would provide insight to policy makers.
